# Dietary Proline Supplementation Promotes Growth and Development in Weaned Foals by Modulating Gut Microbial Amino Acid Metabolism

**DOI:** 10.3390/microorganisms13112598

**Published:** 2025-11-14

**Authors:** Chen Meng, Jianwen Wang, Yaqi Zeng, Xinkui Yao, Jun Meng

**Affiliations:** 1College of Animal Science, Xinjiang Agricultural University, Urumqi 830052, China; chenmeng0330@126.com (C.M.); wjw1262022@126.com (J.W.); zengyaqi@xjau.edu.cn (Y.Z.); yaoxinkui@xjau.edu.cn (X.Y.); 2Xinjiang Key Laboratory of Equine Breeding and Exercise Physiology, Urumqi 830052, China

**Keywords:** proline, weanling foals, fecal microbiota, amino acid metabolism, weaning stress

## Abstract

This study investigated the effects of varying proline supplementation doses in weaned foals. Twenty-eight weaned foals (approximately 5 months of age; body weight: 54.45 ± 11.33 kg; with an equal number of males and females) were randomly assigned to one of four groups—a control group, a low-dose group (20 mg/kg·d), a medium-dose group (40 mg/kg·d), and a high-dose group (60 mg/kg·d)— receiving continuous supplementation for 60 days. Blood samples were collected periodically for the analysis of hormones, antioxidants, immune parameters, and plasma amino acids. Concurrently, fecal 16S rRNA sequencing was performed to assess the microbial community composition. We observed a significant time-dependent interaction between medium-to-high proline supplementation and time. Proline supplementation resulted in dose-dependent increases in foal body weight (*p* = 0.002), hormone levels (*p* < 7.49 × 10^−6^), antioxidant capacity (*p* < 1.56 × 10^−3^), immune function (*p* < 0.005), and key blood biochemical parameters (*p* < 0.019). Concurrently, supplementation with medium and high doses of proline significantly reduced the plasma concentrations of amino acids such as proline and arginine (*p* < 0.05). The medium dose achieved the optimal balance between promoting growth and maintaining high nitrogen utilization efficiency. At the microbial level, medium-dose proline significantly enhanced fecal microbial diversity, particularly enriching characteristic taxa like g_Christensenellaceae_R-7_group. Furthermore, functional inference from PICRUSt2 revealed that medium-dose proline supplementation was associated with a higher microbial potential for amino acid degradation and metabolism, in line with the decreasing plasma concentrations of the corresponding amino acids. This research enhances our understanding of the “host–microbiota interaction” mechanism in weaned foals and provides important theoretical support for mitigating weaning stress and optimizing nutritional strategies.

## 1. Introduction

The weaning period for foals, typically occurring between 4 and 6 months of age, represents a critical turning point and a significant multifaceted stressor. Separation from the mare not only induces psychological anxiety and distress in the foal [1] but also presents severe nutritional and immunological challenges. Weaned foals must rapidly adapt to solid feed, which places immense pressure on their immature digestive systems for nutrient digestion and absorption. Simultaneously, the discontinuation of immune factors provided via colostrum leaves their developing immune systems directly exposed to various pathogens. These compounding stressors severely constrain a foal’s health status and growth potential. Therefore, exploring effective nutritional intervention strategies to mitigate weaning stress and promote a healthy transition is of critical theoretical and practical significance.

Among numerous nutritional regulatory approaches, the application of functional amino acids has emerged as a key research area. Amino acids serve not only as the fundamental building blocks for protein synthesis but also as crucial precursors for numerous bioactive molecules (e.g., polyamines, hormones, neurotransmitters), playing central roles in regulating metabolism, signaling pathways, and overall health [2,3,4]. Historically, the nutrition community held that non-essential amino acids (NEAAs) could be adequately synthesized by the body without requiring supplemental intake. However, mounting evidence challenges this traditional view, indicating that during specific physiological phases—such as stress, disease, or rapid growth—the endogenous synthesis rates of certain NEAAs (e.g., proline, glycine) fail to meet bodily demands [5,6]. This deficit renders them conditionally essential amino acids. As regulators of key metabolic pathways, these amino acids profoundly influence vital processes, including gene expression, nutrient metabolism, and oxidative defense.

Proline exemplifies the class of conditionally essential amino acids [7]. As the only secondary amino acid involved in protein synthesis, it functions not only as a key structural component of collagen but also plays indispensable roles in alleviating stress in young animals [5], regulating cellular energy metabolism [8], and modulating antioxidant [9] and immune responses [10]. Crucially, young animals (such as donkeys, horses, pigs, and sheep) often experience marked weaning stress following abrupt weaning, which is typically manifested by reduced feed intake and diarrhea [11,12,13,14]. During this period of rapid growth and development, their demand for amino acids increases substantially. However, fluctuations in feed intake and intestinal dysfunction after weaning often result in insufficient amino acid intake and absorption [15,16,17,18,19,20]. Therefore, although the body possesses the ability to synthesize proline endogenously, the rate of synthesis may be inadequate to meet the elevated demand associated with stress and rapid growth, leading to an imbalance between supply and demand [5,6]. This imbalance makes dietary proline supplementation an important strategy for maintaining normal growth and physiological function in young animals.

Based on these findings, the nutritional value of proline in promoting growth and development has been confirmed in young animals such as piglets and dairy goats. However, its potential application and underlying mechanisms in weaned foal nutrition remain largely unexplored. This study aims to systematically evaluate the effects of dietary proline supplementation on growth performance, key serum biochemical indicators (amino acids, hormones, antioxidants, and immune factors), and fecal microbiome composition in weaned foals. We hypothesize that proline supplementation can effectively improve the growth status and physiological health of weaned foals. The results of this study will provide direct evidence for revealing the unique role of proline in foal nutrition and offer a scientific basis for optimizing foal feeding management protocols.

## 2. Materials and Methods

### 2.1. Experimental Design

This experiment was conducted at the Zhaosu Horse Farm in Zhaosu County, Xinjiang Uygur Autonomous Region, China. Twenty-eight healthy, weaned foals of the Yili horse breed (approximately 5 months of age; body weight: 54.45 ± 11.33 kg; with an equal number of males and females) were selected. The foals were randomly assigned to one of four treatment groups (*n* = 7 foals per group, comprising four males and three females): Control Group (DG): Fed a basal diet only. Low-Dose Group (LG): Basal diet supplemented with 20 mg/kg·d proline. Medium-Dose Group (MG): Basal diet supplemented with 40 mg/kg·d proline. High-Dose Group (HG): Basal diet supplemented with 60 mg/kg·d proline. The level of proline supplementation was established with reference to our previous findings and prior studies investigating dietary proline supplementation [12]. Proline was thoroughly mixed into the concentrate feed for daily administration. The entire trial included a 14-day acclimation period, followed by a 60-day main experimental period. The composition and nutritional content of the concentrate feed are detailed in Table 1.

During the trial, foals were housed indoors but spent the majority of the day in an outdoor activity area, returning to the barn only during feeding periods. Foals had ad libitum access to water. The total daily concentrate ration (1 kg) was divided into two equal portions and administered via individual feed buckets at 09:00 and 17:00. The specified proline dose for each treatment group was added and mixed into the morning concentrate ration (09:00 feeding). Before feeding, foals were herded into individual stalls for consumption to ensure accurate dosage delivery and subsequently returned to the group activity area.

### 2.2. Sample Collection and Measurement

#### 2.2.1. Foal Body Weight and Body Measurements

Body weight and body size traits were recorded on the mornings before feeding at days 0, 30, and 60 of the trial. Body height and body length (cm) were measured using a measuring stick, chest circumference and cannon bone circumference (cm) were determined with a measuring tape, and body weight (kg) was measured using a floor scale.

#### 2.2.2. Blood Sample Collection and Parameter Measurement in Foals

On days 0 and 60, foals were separated from their dams and fasted for 2 h before blood sampling. Approximately 10 mL of blood was collected from the jugular vein into plain vacuum tubes. Samples were allowed to clot at room temperature for 1 h and centrifuged at 3500 rpm for 15 min. The resulting serum was transferred into 2 mL microtubes, labeled, and stored at −20 °C for the determination of hormonal, immunological, and antioxidant parameters.

Serum biochemical indices, including alanine aminotransferase (ALT), aspartate aminotransferase (AST), total protein (TP), albumin (ALB), triglycerides (TG), glucose (GLU), and blood urea nitrogen (BUN), were analyzed using an automated biochemical analyzer (BA200; colorimetric method).

Antioxidant parameters, including glutathione peroxidase (GSH-PX), malondialdehyde (MDA), superoxide dismutase (SOD), diamine oxidase (DAO), total antioxidant capacity (T-AOC), and catalase (CAT), were determined using a biochemical analyzer (Mindray BS-420; colorimetric method).

Hormonal and immunological indicators, including growth hormone (GH), somatostatin (SS), insulin-like growth factor 1 (IGF-1), immunoglobulin A (IgA), immunoglobulin G (IgG), immunoglobulin M (IgM), interleukin 1β (IL-1β), interleukin 6 (IL-6), interleukin 8 (IL-8), tumor necrosis factor α (TNF-α), and interferon γ (IFN-γ), were quantified by enzyme-linked immunosorbent assay (ELISA) using an enzyme-linked analyzer (Huawei Delong DR-200BS).

On day 60 of the experiment, 10 mL of blood was collected from each foal into heparinized tubes. After processing, plasma samples were aliquoted and stored at −80 °C for targeted amino acid analysis. Plasma amino acids, such as Ala, Arg, Asn, Asp, Cr, Gln, Glu, Gly, His, Ile, Leu, Lys, Met, Orn, Phe, Pro, Ser, Thr, Trp, Tyr, and Val, were quantified using liquid chromatography–tandem mass spectrometry (ExionLC™ AD UHPLC–QTRAP^®^ 6500+) at Novogene Co., Ltd. (Beijing, China).

#### 2.2.3. Collection of Foal Fecal Samples and 16S rRNA Sequencing Analysis

On day 60 of the experiment, fresh rectal fecal samples were collected from all 28 foals. Approximately 5 g of feces per foal was collected, placed in 5 mL cryogenic tubes, immediately flash-frozen in liquid nitrogen, and transferred to −80 °C storage for 16S rRNA sequencing analysis. Sequencing was performed by Novogene Co., Ltd.

Genomic DNA was extracted from fecal samples. The diluted DNA was used as the template for PCR amplification targeting the V3–V4 region of the 16S rRNA gene with specific primers (CCTAYGGGRBGCASCAG and GGACTACHVGGGTWTCTAAT). PCR products were verified by 2.0% agarose gel electrophoresis, purified using magnetic beads, and quantified by a fluorescence-based assay. Equimolar amounts of PCR products were pooled to construct sequencing libraries. After quantification by Qubit fluorometry and qPCR, paired-end sequencing (2 × 250 bp) was performed on the Illumina NovaSeq 6000 platform.

Raw reads were demultiplexed, merged, quality-filtered (to remove low-quality reads and adapter contamination), and checked for chimeras to obtain high-quality sequences (Clean Tags). Sequence denoising and ASV/OTU generation were performed using QIIME2 (version 2022.2). Taxonomic classification was conducted against the SILVA database to generate species abundance tables at different taxonomic levels. α-Diversity indices (Chao1, Shannon, and Simpson) were calculated, and β-diversity was assessed by principal coordinate analysis (PCoA) based on Bray–Curtis distances. Biomarker taxa from phylum to genus levels were identified using the LEfSe (Linear Discriminant Analysis Effect Size) method. Functional predictions of microbial communities were conducted using PICRUSt2 (v2.3.0) based on the KEGG database. Differentially enriched pathways and enzymes were identified using the Wilcoxon rank-sum test, with a screening threshold of |log2 Fold Change| ≥ 1 and *p* < 0.05.

### 2.3. Subsection

All statistical analyses were conducted in R (version 4.4.3). Time-dependent variables, including body measurements and hormonal, antioxidant, and immune parameters, were analyzed using linear mixed-effects models (LMMs). Fixed effects included the proline treatment group (Treatment), sampling time (Time), and their interaction (Treatment × Time), whereas individual foals (ID) were specified as random effects. Linear mixed models were fitted using the lme4 package (version 1.1-37), and *p*-values for fixed effects were obtained using the lmerTest package (version 3.1-3). Estimated marginal means (EMMs) and Tukey’s HSD post hoc comparisons were computed using the emmeans package (version 1.11.0). Multiple comparison adjustments were performed using the multcomp package (version 1.4-28) to control the Type I error rate. Visualization of results was achieved using the ggplot2 (version 3.5.2) and multcompView packages (version 0.1-10).

Pearson correlation analyses were performed in the RStudio (version 4.4.3) environment, with |r| > 0.7 and *p* < 0.05 considered strong correlations. Correlation heatmaps were generated using ggplot2 (version 3.5.2). Intergroup differences in amino acid concentrations, microbial functional pathways, and enzyme abundances were evaluated using one-way ANOVA or Student’s *t*-test in GraphPad Prism (version 9.4.1). Statistical significance was set at ^ns^
*p* > 0.05; * *p* < 0.05; ** *p* < 0.01.

## 3. Results

### 3.1. Dose–Response Patterns of Foal Body Measurements to Different Proline Supplementation Doses

LMM analysis showed that proline supplementation produced a slight, though statistically non-significant, increase in foal height, body length, chest circumference, and cannon bone circumference. The interaction between proline dose and time was not significant for height (Treatment × Time: *F* (6, 48) = 1.72, *p* = 0.13), body length (*F* (6, 48) = 0.48, *p* = 0.84), chest circumference (*F* (6, 48) = 0.43, *p* = 0.86), or cannon bone circumference (*F* (6, 48) = 0.11, *p* = 0.99) (Figure 1A–D; Appendix A).

In contrast, the interaction between proline dose and time had a significant effect on body weight (Treatment × Time: *F* (6, 48) = 4.02, *p* = 0.002). Tukey’s HSD post hoc analysis indicated that the estimated marginal means for the high-dose (166.64 kg; 95% CI: 152.48–180.80) and medium-dose groups (169.29 kg; 95% CI: 155.13–183.45) at day 30 were significantly higher than those of the low-dose (163.43 kg; 95% CI: 149.27–177.59) and control groups (159.14 kg; 95% CI: 144.98–173.30). From day 30 onward, body weight increased markedly, reaching its peak at day 60. By the end of the trial, HG (193.71 kg; 95% CI: 179.55–207.87) and MG (189.29 kg; 95% CI: 175.13–203.45) exhibited significantly greater body weights than LG (178.57 kg; 95% CI: 164.41–192.73) and DG (171.14 kg; 95% CI: 156.98–185.30). These findings demonstrate a clear dose-dependent effect of proline supplementation on body weight, with the high-dose group showing the greatest overall gain in foal growth (Figure 1E; Appendix A).

### 3.2. Dose–Response Patterns of Fetal Hormone Secretion to Different Proline Supplementation Doses

LMM analysis revealed that incorporating proline dose as a continuous variable significantly affected all measured hormonal indices, including GH, S.S., INS, and IGF-1 (Treatment × Time, *p* < 7.5 × 10^−6^), indicating clear dose- and time-dependent changes in endocrine regulation (Appendix A).

Among the hormones, IGF-1 responded most rapidly to proline supplementation, exhibiting a clear positive dose-dependent pattern (Figure 2D). By day 30, IGF-1 concentrations increased significantly with higher proline doses. The estimated marginal mean in the HG was 227.48 ng/mL (95% CI: 219.81–235.15), significantly higher than those in the MG (215.68 ng/mL, 95% CI: 208.01–223.35), LG (205.67 ng/mL, 95% CI: 198.01–213.35), and DG (197.50 ng/mL, 95% CI: 189.83–205.17) groups. This upward trend persisted through day 60, when IGF-1 reached its highest levels. At that time, the HG (241.10 ng/mL, 95% CI: 233.43–248.77) remained significantly higher than MG (225.16 ng/mL, 95% CI: 217.49–232.83), LG (222.42 ng/mL, 95% CI: 214.75–230.09), and DG (197.19 ng/mL, 95% CI: 189.52–204.86) (Appendix A).

The GH response lagged slightly behind IGF-1 but followed a similar positive trajectory (Figure 2A). At day 30, only HG (5.12 ng/mL, 95% CI: 4.81–5.43) differed significantly from DG (4.38 ng/mL, 95% CI: 4.07–4.68), whereas MG and LG showed no significant differences (*p* > 0.05). From day 30 onward, GH concentrations increased progressively with dose, reaching a marked dose-dependent pattern by day 60. The HG (6.79 ng/mL, 95% CI: 6.48–7.10) was significantly higher than MG (5.45 ng/mL, 95% CI: 5.14–5.76), LG (5.09 ng/mL, 95% CI: 4.78–5.40), and DG (4.32 ng/mL, 95% CI: 4.01–4.63). Both MG and LG also differed significantly from DG (*p* < 5.2 × 10^−5^).

In contrast, S.S. and INS exhibited inverse dose-dependent responses (Figure 2B,C). For S.S., the control group showed the highest concentration at day 30 (30.13 pg/mL, 95% CI: 28.63–31.63), which was significantly greater than in all supplementation groups, with HG showing the lowest level (19.13 pg/mL, 95% CI: 17.63–20.64). By day 60, these differences became more pronounced. The DG (30.92 pg/mL, 95% CI: 29.42–32.43) remained significantly higher than all supplementation groups, and LG (25.50 pg/mL) was also higher than HG (23.71 pg/mL), reflecting a sustained, dose-dependent decrease in S.S. levels over time (Appendix A).

### 3.3. Dose–Response Patterns of Antioxidant Indicators in Foals Supplemented with Different Proline Doses

LMM analysis revealed that the interaction between proline dose and time had a highly significant effect on all measured antioxidant parameters (*p* < 1.56 × 10^−3^), indicating consistent dose-dependent responses to both supplementation level and duration (Appendix A).

Following proline supplementation, the response patterns of SOD, GSH-PX, T-AOC, and CAT were generally consistent, showing clear positive dose-dependent trends (Figure 3A–C,F). At day 30, all four antioxidant indices in the supplementation groups were markedly higher than those in the DG, with an emerging dose-dependent pattern particularly evident for GSH-PX and T-AOC. The estimated marginal means of GSH-PX and T-AOC in the HG were 165.41 U/mL (95% CI: 159.63–171.18) and 9.35 U/mL (95% CI: 8.87–9.83), respectively, significantly higher than in the MG (GSH-PX: 157.83 U/mL, 95% CI: 152.05–163.61; T-AOC: 8.70 U/mL, 95% CI: 8.22–9.18), the LG, and the DG (GSH-PX: 140.83 U/mL, 95% CI: 135.06–146.61; T-AOC: 6.87 U/mL, 95% CI: 6.39–7.34).

By day 60, all four parameters reached their peak levels. Dose-dependent responses for GSH-PX and T-AOC were fully established, with the HG showing the highest values among all groups (GSH-PX: 173.94 U/mL, 95% CI: 168.16–179.71; T-AOC: 10.02 U/mL, 95% CI: 9.55–10.50). Significant differences were observed among the supplementation groups (Appendix A).

In contrast, DAO and MDA exhibited consistent negative dose-dependent responses (Figure 3D,E). By day 30, both indicators were significantly lower in all proline-supplemented groups than in the DG (*p* < 1.08 × 10^−3^), although no significant differences were detected among the supplemented groups (*p* > 0.05). By day 60, DAO and MDA levels in the HG had decreased to 3.49 U/mL (95% CI: 3.01–3.97) and 5.62 nmol/mL (95% CI: 5.26–5.97), respectively, representing the lowest levels observed. These findings indicate that proline supplementation effectively mitigated oxidative stress in the body (Appendix A).

### 3.4. Dose–Response Patterns of Immunological Indicators in Foals Supplemented with Different Proline Doses

LMM analysis revealed a highly significant interaction between proline dose and time for TNF-α, IFN-γ, IL-1β, IgG, and IgM (*p* < 0.005), indicating that these immunological indicators exhibited dose-dependent changes with both supplementation level and duration (Appendix A).

All three proinflammatory cytokines (TNF-α, IFN-γ, and IL-1β) showed clear negative dose-dependent responses (Figure 4A–C). Among them, IL-1β responded most rapidly to proline supplementation, with a negative dose-dependent pattern already established by day 30. The estimated marginal mean in the DG was 17.87 pg/mL (95% CI: 17.26–18.48), significantly higher than in all supplementation groups (*p* < 2.56 × 10^−8^). IL-1β levels differed significantly among the supplemented groups (*p* < 4.29 × 10^−2^), with the HG showing the lowest level (13.55 pg/mL, 95% CI: 12.94–14.16). By day 60, concentrations reached their minimum values, and the DG (17.50 pg/mL, 95% CI: 16.89–18.11) remained significantly higher than the LG (14.43 pg/mL, 95% CI: 13.82–15.04), MG (13.54 pg/mL, 95% CI: 12.93–14.15), and HG (12.66 pg/mL, 95% CI: 12.05–13.28) groups (Appendix A).

TNF-α and IFN-γ followed similar trends but responded more slowly than IL-1β (Figure 4A,B). For TNF-α, the dose-dependent pattern was not fully established at day 30 and became evident only toward the end of the trial (Appendix A).

IgM and IgG displayed comparable overall response trends, although their dose-dependent patterns differed (Figure 4H,I). For IgG, concentrations increased steadily in all supplemented groups during the first 30 days, while remaining unchanged in the control group. After day 30, divergent trends emerged: IgG levels in the LG continued to rise, peaking at 22.72 g/L (95% CI: 20.01–25.43) at day 60, whereas those in the MG and HG began to decline, reaching 16.02 g/L (95% CI: 13.31–18.72) in the HG by day 60. The DG maintained nearly constant IgG concentrations throughout the study. IgM showed a similar time-course pattern to IgG (Figure 4I), indicating that low-dose supplementation promoted sustained immunoglobulin synthesis, whereas medium- and high-dose treatments tended to suppress it during the later stages of the trial (Appendix A).

### 3.5. Dose–Response Patterns of Biochemical Indicators in Foals to Different Proline Supplementation Doses

LMM analysis revealed a significant interaction between proline dose and time for AST, ALT, and BUN (*p* < 0.019), indicating dose-dependent changes with both supplementation level and duration (Appendix A).

The response patterns of the two liver enzyme indicators (AST and ALT) were largely consistent (Figure 5A,C), showing clear dose-dependent increases. By day 30, the HG showed a markedly higher AST level than the DG (*p* < 0.0001), while no significant differences were observed among the other groups (*p* > 0.05). The estimated marginal mean for the HG was 287.83 U/L (95% CI: 275.95–299.71), compared with 259.47 U/L (95% CI: 247.59–271.35) in the DG, and this difference remained evident throughout the experimental period.

For ALT, levels in all proline-supplemented groups (LG, MG, and HG) were significantly higher than those in the control group by day 30 (*p* < 0.004), although no significant differences were observed among the supplementation groups (*p* > 0.05). At day 60, the estimated marginal means were as follows: HG, 14.00 U/L (95% CI: 11.62–15.05); MG, 13.67 U/L (95% CI: 11.95–15.38); LG, 13.33 U/L (95% CI: 11.62–15.05); and DG, 11.00 U/L (95% CI: 9.28–12.72) (Appendix A). BUN levels exhibited a downward trend with increasing proline supplementation. By day 30, BUN concentrations were significantly higher than those in the DG (*p* < 0.01) in all supplementation groups except LG (*p* > 0.16). By day 60, BUN levels had increased in all supplementation groups, with the HG showing significantly higher values than the other three groups (*p* < 0.02) (Appendix A).

### 3.6. Effects of Proline Supplementation on Plasma Free Amino Acid Abundance in Foals

Different doses of proline supplementation significantly affected plasma free amino acid profiles in foals. Compared with the control group, medium-dose proline supplementation markedly decreased the plasma concentrations of isoleucine, leucine, serine, threonine, proline, and arginine (*p* < 0.05) (Figure 6A). In contrast, high-dose proline supplementation significantly decreased plasma proline and arginine levels (*p* < 0.05) (Figure 6B).

Differentially enriched amino acids were mainly involved in several amino acid metabolic pathways, including arginine and proline metabolism, glycine, serine, and threonine metabolism, and valine, leucine, and isoleucine degradation (*p* < 0.05). These amino acids also participated in pathways associated with cofactor and vitamin metabolism (one-carbon pool by folate, pantothenate and CoA biosynthesis), other amino acid metabolism (glutathione metabolism), and central carbon metabolism (glyoxylate and dicarboxylate metabolism) (*p* < 0.05) (Figure 6C).

### 3.7. Structural and Richness Responses of Foal Fecal Microbiota Following Different Doses of Proline Supplementation

Analysis of the fecal microbiome composition in foals revealed that Firmicutes and Bacteroidetes were the dominant phyla (Figure 7A). At the family level, the predominant taxa were Lachnospiraceae, F082, Oscillospiraceae, Methanobacteriaceae, and Rikenellaceae (Figure 7B). At the genus level, the major genera included UCG-002, Methanobrevibacter, Rikenellaceae_RC9_gut_group, Akkermansia, NK4A214_group, and Christensenellaceae_R-7_group (Figure 7C). These results characterized the baseline microbial community composition across all samples.

To further elucidate how proline supplementation modulated microbial community structure, relative abundance heatmaps were generated at the phylum, family, and genus levels. At the phylum level, Firmicutes and Bacteroidetes showed the most pronounced responses to proline supplementation, with Firmicutes displaying a marked increase in relative abundance in the MG (Figure 7D). At the family level, Christensenellaceae, F082, and Lachnospiraceae were significantly enriched in the MG, whereas Oscillospiraceae was more abundant in the DG and HG (Figure 7E). At the genus level, proline supplementation decreased the relative abundance of UCG-002 and Methanobrevibacter while increasing that of Christensenellaceae_R-7_group (Figure 7F).

Good’s coverage exceeded 0.995 in all samples, indicating sufficient sequencing depth to capture the majority of microbial taxa (Figure 7G). Alpha diversity was evaluated using the Chao1 index (species richness), Shannon index (diversity), and Simpson index (evenness). The MG showed significantly higher Chao1-based species richness than the DG and LG (Figure 7H), whereas no significant differences were detected among groups for the Shannon or Simpson indices (Figure 7I,J). Principal coordinate analysis (PCoA) based on Bray–Curtis distances showed that PCoA1 and PCoA2 explained 9.69% and 10.41% of the total variance, respectively. The microbial community structure differed significantly between the DG and MG (Adonis: R^2^ = 0.17, *p* = 0.042), while no significant differences were observed among the other groups (*p* > 0.05) (Figure 7K).

Further LEfSe analysis (LDA > 3.5) identified discriminant taxa among groups. The DG was characterized by c_Clostridia, whereas the MG was enriched in g_Christensenellaceae_R-7_group, f_Christensenellaceae, and o_Christensenellales (Figure 7L,M). No discriminant taxa were detected in the LG and HG. The corresponding heatmap showed higher relative abundance of c_Clostridia in DG and MG, while f_Christensenellaceae and o_Christensenellales exhibited the greatest enrichment in MG. Although g_Christensenellaceae_R-7_group showed modest variation across all treatments, it remained most abundant in MG (Figure 7N).

### 3.8. Functional Prediction of Foal Fecal Microbiota

To investigate the impact of proline supplementation on the functional potential of foal gut microbiota, we performed KEGG pathway prediction on 16S rRNA sequencing data using PICRUSt2 (version 2.6.0). Compared to the DG, the MG exhibited significantly enhanced functional potential across multiple metabolic pathways in fecal microbiota. These pathways primarily clustered around amino acid metabolism, including arginine and proline metabolism, valine, leucine and isoleucine metabolism, and glycine, serine and threonine metabolism (*p* < 0.05). Additionally, predicted abundances for pathways such as Steroid hormone biosynthesis, Lipopolysaccharide biosynthesis, Secondary bile acid biosynthesis, and glycolysis/gluconeogenesis were significantly higher in the MG than in the DG (*p* < 0.05) (Figure 8A). To identify the differential enzymes driving these pathway changes, we further compared predicted enzyme abundances. Results revealed multiple differentially expressed enzymes significantly upregulated in the MG, including amino acid degrading enzymes involved in valine, leucine and isoleucine metabolism (fadB, gshB, and EC:5.4.99.2); amino acid degrading enzymes associated with arginine and proline metabolism (hyuA, hyuB, and rocF); amino acid degrading enzymes related to glycine, serine and threonine metabolism (ectB); while amino acid synthase (metL) was significantly downregulated; enzymes related to glycolysis/gluconeogenesis such as aceE and agp; and enzymes related to Steroid hormone biosynthesis such as AKR1C4 and fabG3 (*p* < 0.05) (Figure 8A).

A similar pattern emerged in the comparison between the HG and DG, with significantly enhanced functional potential observed in amino acid metabolism (arginine and proline metabolism, valine, leucine, and isoleucine biosynthesis, glycine, serine and threonine metabolism) and energy metabolism (TCA cycle, butanoate metabolism, glycolysis/gluconeogenesis) showed significantly enhanced functional potential, with corresponding increases in predicted abundance of key enzymes associated with energy metabolism and amino acid catabolism (*p* < 0.05) (Figure 8B). Interestingly, although total abundance differed between HG and LG in certain pathways, the predicted abundance of most enzymes within these pathways showed no significant difference (*p* > 0.05) (Figure 8C).

Finally, Pearson correlation analysis was performed between predicted functional abundance of the microbiota and plasma amino acid concentrations. Results indicated that in both MG and DG, fecal microbiota abundance in the arginine and proline metabolic pathways showed significant negative correlations with serum proline and arginine concentrations (|r| > 0.60, *p* < 0.05) (Figure 8D).

## 4. Discussion

Weaning represents a critical stress period in the life of young foals, accompanied by multifaceted physiological and psychological challenges that can compromise both nutritional status and immune function [1]. Under such stress, the endogenous synthesis of certain non-essential amino acids often fails to meet physiological demands, rendering them conditionally essential [5]. For example, livestock have limited capacity to synthesize sufficient glutamine under stress, necessitating dietary supplementation. Similarly, proline is a key structural component of collagen and elastin and is among the amino acids with the highest systemic protein synthesis requirements. Its content in plant-based feeds is typically low [15]. Therefore, exogenous proline supplementation is essential to support growth and development during the weaning period.

The results of this study demonstrate that dietary proline supplementation after weaning effectively improves growth performance, antioxidant capacity, and immune function in foals, with a significant dose × time interaction. Specifically, a moderate dose (40 mg/kg·d) appears to represent an effective threshold, while higher doses exert more pronounced growth-promoting effects. Among the measured parameters, body weight change—a core indicator of growth status—directly reflected the effectiveness of feeding management. From day 30 onward, foals in the medium- and high-dose groups exhibited significant, dose-dependent increases in body weight, which persisted until the end of the trial. These findings provide direct evidence that proline supplementation promotes foal growth and development. Previous studies have similarly reported that proline enhances growth and development in young livestock [11,12,13,14], supporting our observations.

Growing evidence indicates that proline functions not only as a substrate for protein synthesis but also as a signaling molecule and endocrine regulator, playing a pivotal role in stimulating growth hormone (GH) and insulin-like growth factor 1 (IGF-1) secretion [21]. Consistent with this, foals receiving moderate-to-high doses of proline in this study showed marked, dose-dependent increases in serum GH and IGF-1 levels. GH [21,22,23] and IGF-1 [21,24,25] are key hormones promoting growth in young animals. Their synergistic action enhances tissue perfusion and amino acid transport to the skeletal muscle, thereby supporting protein synthesis and muscle growth. In contrast, somatostatin (S.S.) and insulin (INS) levels decreased in a dose-dependent manner with increasing proline supplementation. Collectively, the activation of GH and IGF-1, along with reduced S.S., likely contributed to maintaining robust growth momentum in foals during the weaning period.

Weaning stress induces physiological and metabolic disturbances that typically reduce antioxidant capacity and increase the risk of oxidative damage in intestinal tissues [26]. In the present study, dietary proline supplementation dose-dependently enhanced serum superoxide dismutase (SOD), glutathione peroxidase (GSH-PX), total antioxidant capacity (T-AOC), and catalase (CAT) activity in foals, indicating that proline strengthens the endogenous antioxidant defense system. Oxidative stress also triggers lipid peroxidation, generating malondialdehyde (MDA) [27], while impaired intestinal barriers allow diamine oxidase (DAO) leakage into the bloodstream, with serum DAO levels reflecting intestinal injury severity [28,29]. Proline supplementation significantly reduced serum MDA and DAO concentrations in a dose-dependent manner, suggesting effective alleviation of oxidative stress and maintenance of intestinal barrier integrity. This protective effect may result from proline’s capacity to scavenge reactive oxygen species and its role as a substrate for citrulline and arginine synthesis, which subsequently generates nitric oxide (NO) to further neutralize free radicals [30,31,32].

Proline also plays a critical role in immune cell metabolism, proliferation, differentiation, and regulation of inflammatory mediators [33]. In this study, all proline-supplemented groups exhibited significantly elevated serum IgG, IgA, and IgM levels compared to controls. Although the immunoglobulin-promoting effect was similar across doses, even low-dose supplementation effectively enhanced humoral immunity. Concurrently, stress-induced release of proinflammatory factors (TNF-α, IFN-γ, IL-1β) diverts nutrients toward immune responses, impairing growth [34,35,36]. Medium- and high-dose proline supplementation significantly inhibited these proinflammatory cytokines in a dose-dependent manner, highlighting proline’s immunomodulatory potential in alleviating immune stress and maintaining efficient nutrient utilization.

Proline supplementation also markedly influenced systemic amino acid metabolism. Both medium and high doses reduced plasma concentrations of several amino acids, including arginine, proline, and branched-chain amino acids (BCAAs: Leu, Ile). Contrary to expectations, long-term proline supplementation did not increase plasma proline levels and instead led to reduced abundance of multiple amino acids, indicating a substantial remodeling of amino acid metabolism. This may result from accelerated tissue uptake driven by elevated GH and IGF-1 during rapid growth, which increases amino acid transport to skeletal muscle and supports protein synthesis [21,22,23,24,25]. Leucine further promotes protein synthesis via mTOR signaling and enhances BCAA catabolism through BCKDH activation [37,38,39,40]. Furthermore, proline catabolism may enhance arginine synthesis, which in turn induces arginase expression, activates the urea cycle, and accelerates amino acid catabolism [41,42,43,44]. This metabolic shift may explain several observations in the HG foals: on one hand, the early increase in IgG and IgM levels may be related to enhanced amino acid turnover, providing sufficient substrates for immunoglobulin synthesis; on the other hand, the concomitant elevation of plasma urea nitrogen (BUN) directly reflects increased amino acid catabolism and nitrogen metabolism, suggesting a shift in metabolic state rather than a simple improvement in utilization efficiency [45]. Notably, immunoglobulin synthesis incurs substantial metabolic costs, and the subsequent decline in immunoglobulin levels may reflect a trade-off, wherein resources are preferentially allocated to rapid growth rather than the sustained production of high levels of immune proteins [46]. Overall, for weanling foals, a supplementation dose of 60 mg/kg·d appears to promote growth but may approach the upper limit of the optimal metabolic range. Future studies should incorporate more comprehensive clinical chemistry and histopathological assessments to further evaluate its long-term safety. In contrast, the MG achieved effective growth while maintaining lower BUN levels, suggesting that this dose may achieve a more favorable balance among substrate supply, nitrogen metabolism, and immune homeostasis.

Given that reduced plasma amino acid levels cannot be fully explained by tissue uptake alone, we further examined gut microbiota. Recent studies highlight the critical role of gut microbial communities in maintaining host amino acid homeostasis [47]. Dietary proline supplementation significantly altered the structure and richness of the foal gut microbiota, suggesting that microbial metabolic remodeling is another important driver of the observed changes in plasma amino acid levels. Escherichia coli exhibits high amino acid degradation rates, preferring lysine, arginine, proline, and BCAAs (leucine, valine, isoleucine) as substrates [48,49,50,51]. This affects host intestinal epithelial signaling and regulates bacterial gene expression [52,53], thereby modulating the production of enzymes involved in amino acid catabolism [54,55].

Functional prediction analysis using PICRUSt2 (version 2.6.0) revealed that medium-to-high proline supplementation significantly enhanced amino acid catabolic potential, including branched-chain amino acids, in foal fecal microbiota. Specifically, compared to the control group, the MG and HG exhibited significantly higher predicted abundances of pathways involved in arginine and proline metabolism, valine, leucine, and isoleucine metabolism, as well as glycine, serine, and threonine metabolism. Key amino acid-degrading enzymes, including fadB, gshB, hyuA, hyuB, rocF, and ectB, were also upregulated. Importantly, the abundance of microbial pathways involved in amino acid metabolism was negatively correlated with the corresponding amino acid concentrations in host plasma, suggesting that remodeling of gut microbial amino acid catabolism may play a key role in regulating systemic amino acid homeostasis. In addition, proline supplementation increased the predicted abundance of short-chain fatty acid (SCFA) synthesis pathways, particularly butyrate, in foal microbiota. SCFAs serve as the primary energy source for intestinal epithelial cells and play critical roles in maintaining intestinal barrier integrity and modulating immune and inflammatory responses [56,57,58]. This enhancement may provide additional metabolic and gut health benefits for weaned foals.

Overall, this study systematically elucidates the mechanistic pathways by which proline supplementation promotes growth in weaned foals, offering both theoretical insights into foal lactation physiology and practical guidance for reducing weaning stress. Limitations exist, including a relatively small sample size and the inherent constraints of functional inference based on 16S rRNA sequencing and PICRUSt2. Future studies should explore a finer proline dose gradient to determine the optimal supplementation level and validate causal microbial functions using fecal microbiota transplantation (FMT). Integrating precise quantification of key metabolites (e.g., SCFAs, TCA cycle intermediates, hormones) with measurements of relevant enzyme activities will further confirm the molecular mechanisms underlying proline’s effects on growth and metabolism in weaned foals.

## 5. Conclusions

This study investigated how dietary proline supplementation influences the growth of weaned foals, with a particular focus on the potential interplay between the gut microbiota and host amino acid metabolism. Our results indicate that a moderate dose of proline (40 mg/kg·d) achieved an optimal balance between growth promotion and nitrogen utilization under the experimental conditions. Notably, proline supplementation was associated with predicted remodeling of microbial amino acid catabolic functions, and these inferred microbial changes were inversely correlated with plasma amino acid concentrations. This correlative evidence suggests that microbiota-mediated modulation may be one of the mechanisms through which proline affects growth. Furthermore, our data support the notion that proline contributes to growth by influencing growth-related hormones, immune function, and antioxidant capacity. Collectively, these findings provide initial insights into host–microbiota interactions during weaning and offer a foundational framework for developing nutritional strategies to mitigate weaning stress. Future studies, including causal experiments such as fecal microbiota transplantation and investigations across other breeds and developmental stages, are warranted to confirm the generalizability and mechanistic basis of these observations.

## Figures and Tables

**Figure 1 microorganisms-13-02598-f001:**
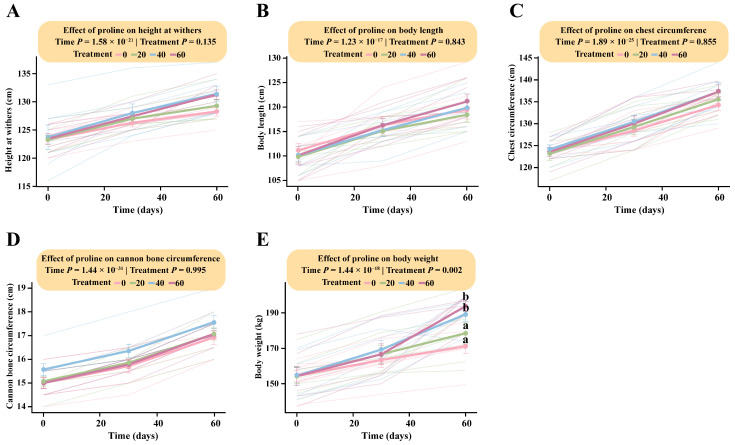
Dose-dependent responses of foal body measurements to different levels of proline supplementation. The interaction between proline treatment and time was analyzed using a linear mixed-effects model (LMM). Pairwise comparisons were performed using Tukey’s HSD test. Data points sharing the same letter are not significantly different (*p* > 0.05); and different lowercase letters indicate significant differences (*p* < 0.05). (**A**) Body height. (**B**) Body length. (**C**) Chest circumference. (**D**) Cannon bone circumference. (**E**) Body weight.

**Figure 2 microorganisms-13-02598-f002:**
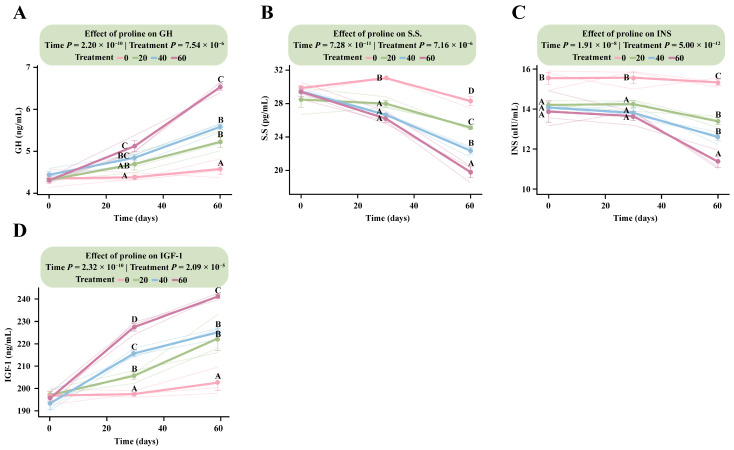
Dose–response patterns of fetal hormone secretion to different proline supplementation doses. The interaction effects between proline treatment (Treatment) and time (Time) were analyzed using a linear mixed model. Data within figures underwent Tukey HSD post hoc testing for multiple comparisons. Data points sharing the same letter are not significantly different (*p* > 0.05); and different uppercase letters indicate highly significant differences (*p* < 0.01). (**A**) Growth hormone. (**B**) Somatostatin. (**C**) Insulin. (**D**) Insulin-like growth factor-1.

**Figure 3 microorganisms-13-02598-f003:**
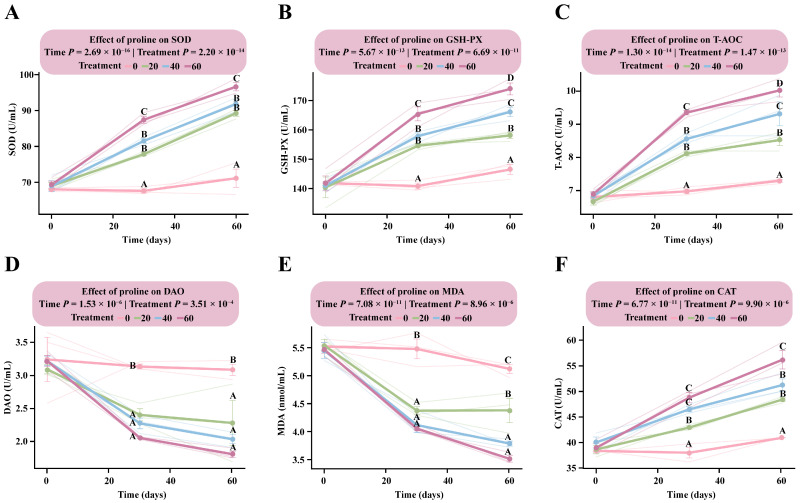
Dose–response patterns of antioxidant indicators in foals supplemented with different levels of proline. The interaction between proline dose and time was analyzed using a linear mixed model. Multiple comparisons within each figure were conducted using Tukey’s HSD test. Bars sharing the same letter are not significantly different (*p* > 0.05); and different uppercase letters denote highly significant differences (*p* < 0.01). (**A**) Superoxide dismutase. (**B**) Glutathione peroxidase. (**C**) Total antioxidant capacity. (**D**) Diamine oxidase. (**E**) Malondialdehyde. (**F**) Catalase.

**Figure 4 microorganisms-13-02598-f004:**
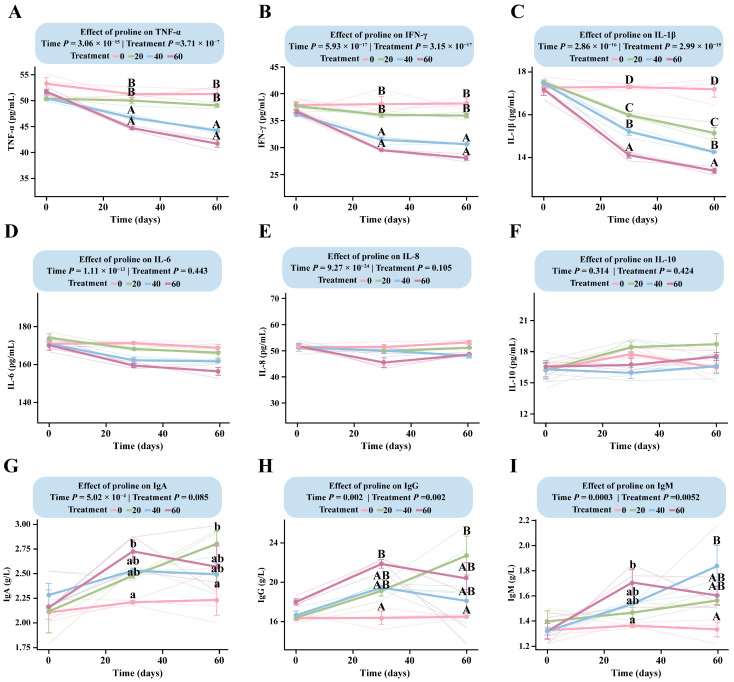
Dose–response patterns of immune indicators in foals supplemented with different levels of proline. The interaction between proline dose and time was analyzed using a linear mixed-effects model. Multiple comparisons within each figure were performed using Tukey’s HSD post hoc test. Bars sharing the same letter are not significantly different (*p* > 0.05); different lowercase letters indicate significant differences (*p* < 0.05); and different uppercase letters denote highly significant differences (*p* < 0.01). (**A**) TNF-α. (**B**) IFN-γ. (**C**) IL-1β. (**D**) IL-6. (**E**) IL-8. (**F**) IL-10. (**G**) IgA. (**H**) IgG. (**I**) IgM.

**Figure 5 microorganisms-13-02598-f005:**
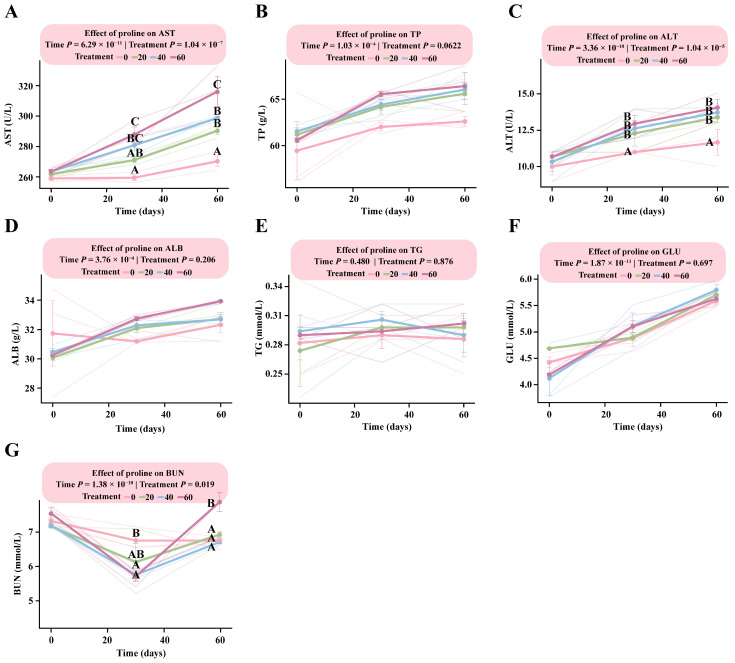
Dose–response patterns of biochemical indicators in foals supplemented with different levels of proline. The interaction between proline dose and time was analyzed using a linear mixed-effects model. Multiple comparisons within each figure were performed using Tukey’s HSD post hoc test. Bars sharing the same letter are not significantly different (*p* > 0.05); and different uppercase letters denote highly significant differences (*p* < 0.01). (**A**) Aspartate aminotransferase. (**B**) Total protein. (**C**) Alanine aminotransferase. (**D**) Albumin. (**E**) Triglycerides. (**F**) Glucose. (**G**) Blood urea nitrogen.

**Figure 6 microorganisms-13-02598-f006:**
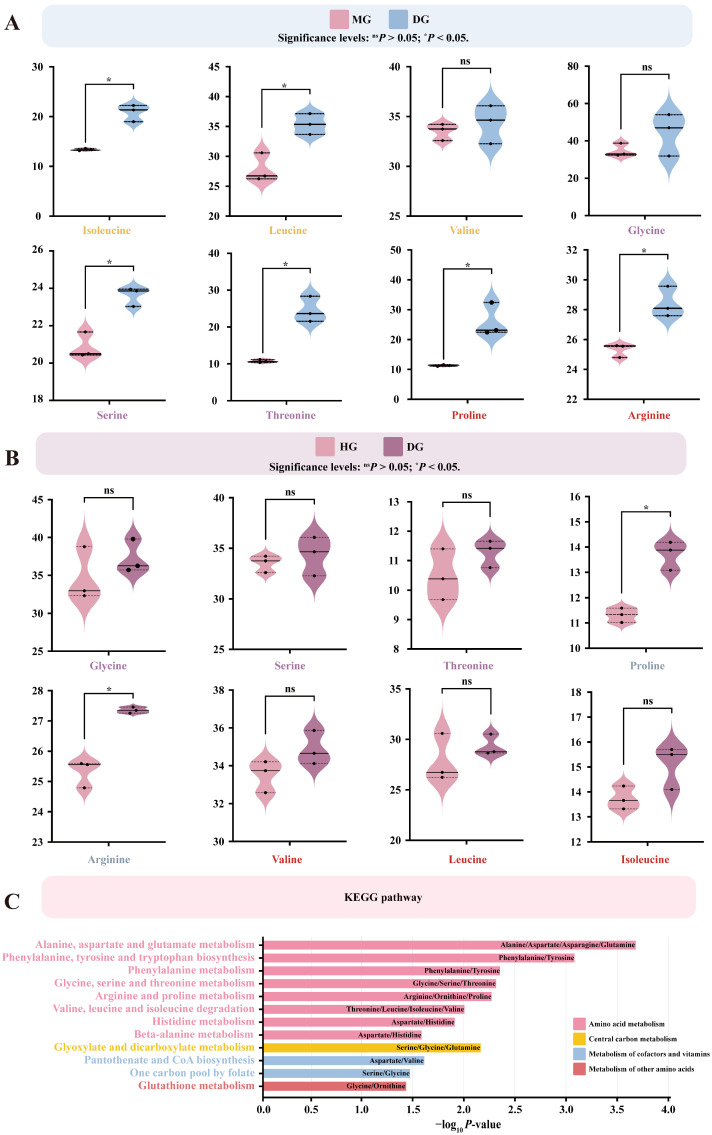
Effects of proline supplementation on plasma free amino acid abundance in foals. Differential amino acids were identified based on linear model analysis followed by Tukey’s HSD post hoc test (*p* < 0.05). Pathway enrichment was performed using KEGG annotation. (**A**) Differentially expressed amino acids in the MG compared with the DG. (**B**) Differentially expressed amino acids in the HG compared with the DG. (**C**) KEGG-enriched metabolic pathways of differentially expressed amino acids.

**Figure 7 microorganisms-13-02598-f007:**
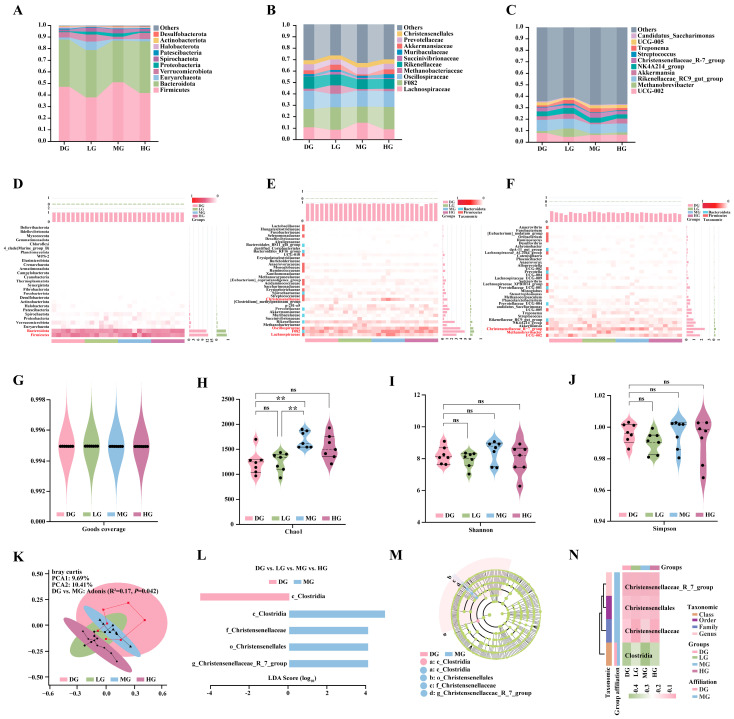
Effects of different proline supplementation doses on the structure and diversity of foal fecal microbiota. (**A**) Phylum-level community composition. (**B**) Family-level community composition. (**C**) Genus-level community composition. (**D**) Heatmap of phylum-level relative abundance. (**E**) Heatmap of family-level relative abundance. (**F**) Heatmap of genus-level relative abundance. (**G**) Sequencing depth assessed by Good’s coverage across groups. (**H**) α-diversity (Chao1 index) of fecal microbiota across groups. (**I**) α-diversity (Shannon index) of fecal microbiota across groups. (**J**) α-diversity (Simpson index) of fecal microbiota across groups. (**K**) Principal coordinate analysis (PCoA) based on Bray–Curtis distances. (**L**) LEfSe analysis showing LDA scores of discriminant taxa. (**M**) Phylogenetic tree of discriminant taxa. (**N**) Heatmap of relative abundance differences for discriminant taxa across groups. Statistical significance for panels D–G was determined by one-way ANOVA. ^ns^
*p* > 0.05; ** *p* < 0.01.

**Figure 8 microorganisms-13-02598-f008:**
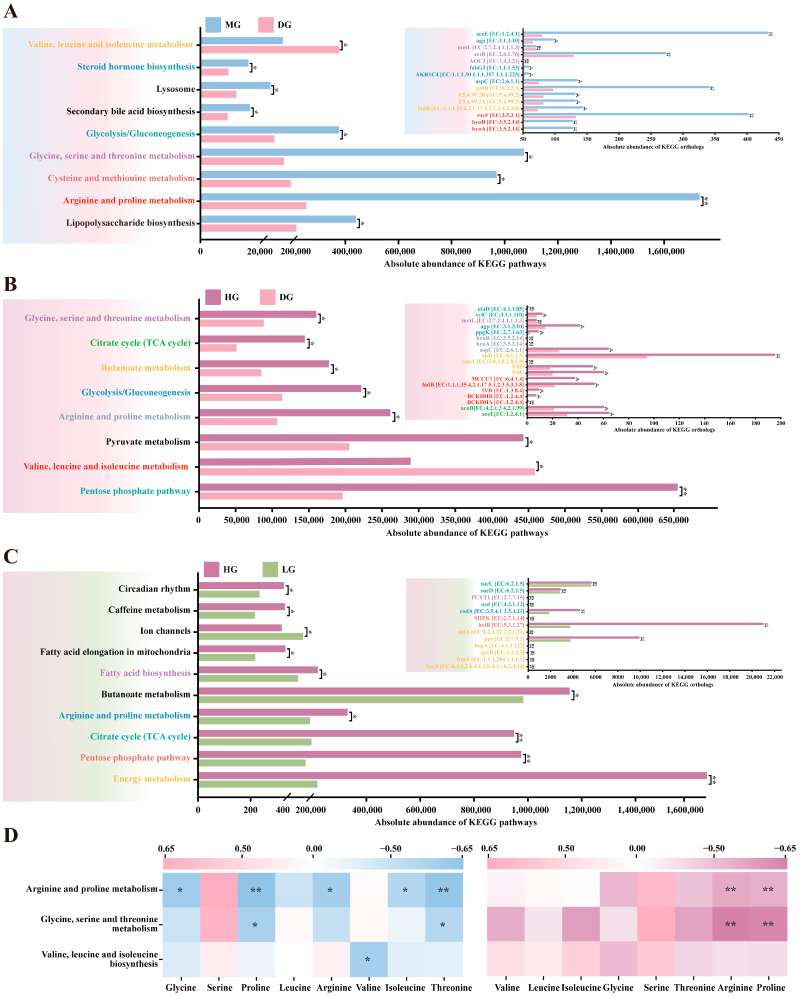
KEGG functional prediction of weaned foal fecal microbiota based on PICRUSt2. (**A**) Differences in KEGG pathways and predicted enzyme (ortholog) abundances between MG and DG. (**B**) Differences in KEGG pathways and predicted enzyme abundances between HG and DG. (**C**) Differences in KEGG pathways and predicted enzyme abundances between HG and LG. (**D**) Pearson correlation heatmap between microbial amino acid metabolic pathways and plasma amino acid concentrations. Statistical significance for panels (**A**–**C**) was assessed using Student’s *t*-test. ^ns^
*p* > 0.05; * *p* < 0.05; ** *p* < 0.01.

**Table 1 microorganisms-13-02598-t001:** Composition and nutrient level of concentrate supplement (DM basis).

Ingredients	Content (%)	Nutrient Components ^2^	Content (%)
Corn	46	DM	90.76
Barley	8	CP	14.44
Wheat bran	8	NDF	61.68
Rapeseed meal	28	ADF	9.95
Premix ^1^	10	Ca	0.63
Total	100	P	0.35

^1^ Premix provided per kilogram of concentrate: Vitamin A 120,000 IU; Vitamin D_3_ 30,000 IU; Vitamin E 2500 IU; Niacin 300 mg; Copper 250 mg; Iron 1200 mg; Zinc 1200 mg; Manganese 1100 mg; Iodine 8 mg; Selenium 6 mg; Cobalt 4 mg. ^2^ Nutrient contents are measured values.

## Data Availability

Raw reads of Transcriptomic sequencing of blood are available at CNCB. GSA submission information: CRA032040. https://ngdc.cncb.ac.cn/gsa/browse/CRA032040 (accessed on 27 October 2025).

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
