# Peer review of "Dietary Proline Supplementation Promotes Growth and Development in Weaned Foals by Modulating Gut Microbial Amino Acid Metabolism"

_microorganisms, 2025, doi:10.3390/microorganisms13112598_

Round 1
Reviewer 1 Report
Comments and Suggestions for Authors
Title: “Dietary Proline Supplementation Promotes Growth and Development in Weaned Foals by Remodeling Intestinal Amino Acid Metabolism”
Dear Authors,
The study addresses an interesting topic within the field of equine nutrition and offers an innovative approach to the role of proline in the development of weaned foals. However, after a detailed analysis of the manuscript, several methodological and interpretative weaknesses have been identified that should be addressed before the paper can be considered for publication. The following observations are offered in a constructive spirit, with the aim of strengthening the scientific validity and clarity of the work.
Title and Abstract
The title should be improved: it promises a mechanistic demonstration (“remodeling intestinal amino acid metabolism”), yet the study only includes functional inferences derived from PICRUSt2 and no direct metabolic evidence (e.g., metabolomics, isotopic flux or actual enzyme measurement).
The abstract contains causal statements (“enhances microbial amino acid degradation capacity”) without direct experimental support; the study merely predicts functional pathways without in vivo validation.
There is a lack of clarity regarding the actual number of animals included after the adaptation period (28 or 24 foals); this methodological inconsistency affects statistical credibility.
The sex of the animals is not mentioned as a covariate, nor is the initial homogeneity between groups (weight, age), which limits interpretation of the “dose–time effect.”
Was the reduction of plasma amino acids evaluated as a consequence of supplementation or as an outcome of physiological variability among individuals?
Introduction
Although well written, the introduction relies excessively on general references (Wu et al., 2013; Hou & Wu, 2018) and extrapolates evidence from other species (pigs, goats) without justifying its physiological applicability to Equus caballus.
The physiological basis of the proline requirement in foals is not identified, nor are the baseline levels in the control diet, making it impossible to determine whether the doses used are supplementary or excessive.
Is there literature supporting the selected dosage (20–60 mg/kg·d) based on equine metabolism, or was it merely extrapolated from non-equine monogastrics?
Materials and Methods
The description of the diet lacks data on metabolizable energy, effective fibre and lysine:proline ratio; without these, it is impossible to assess nutritional balance.
The sample size (n = 7) is small for applying linear mixed models with multiple interactions (Treatment × Time × Variable); this increases the risk of type I errors.
No indication is given of individual feed intake control; since animals were fed in groups, it is uncertain whether each foal actually received the prescribed dose.
The microbiome analysis is based exclusively on 16S rRNA without validation by qPCR of differential taxa or verification of functional abundance through metatranscriptomics?.
Were confounding factors such as antibiotic use, parasitism, or forage differences controlled, which might have affected the faecal microbiome?
Results
Numerous multiple comparisons are presented with extremely low p values (10⁻⁶ to 10⁻³) without reporting effect sizes or interpretable confidence intervals.
Statistical significance is conflated with biological relevance; for instance, marginal increases in AST and ALT are interpreted as “metabolic improvements,” although they may indicate hepatic overload.
Plasma amino acid results are interpreted as “efficiency of utilisation,” yet their decrease may stem from excessive catabolism or tissue redistribution, not from metabolic optimisation.
In microbiota data, differences in α-diversity (Chao1, Shannon) are minimal and could result from interindividual variability; no rarefaction validation or covariance analysis is presented.
Was intra-individual consistency of microbial profiles (before vs after the trial) evaluated, or were only cross-sectional comparisons made at the end of the experiment?
Discussion
The causal link between supplementation and microbial remodelling is overinterpreted; the evidence derives from functional inference, not from direct bacterial metabolic measurements.
Proline is cited as a modulator of the GH/IGF-1 axis, but the references (Maimaituxun et al., 2024; Yan et al., 2024) refer to humans or unrelated physiological contexts (thermoregulation, cardioprotection).
Possible adverse effects of high doses (60 mg/kg·d) are not discussed, despite increased AST, ALT and BUN values—indicators of hepatic stress or excessive catabolism.
A critical analysis of the limitations of PICRUSt2 and its precision in inferring functions in equine microbiota (not represented in databases) is missing.
The discussion fails to compare immunoglobulin responses with the potential metabolic cost of increased serum proteins and does not address temporal dynamics (decrease in IgG and IgM at high doses).
Could the reduction in plasma amino acids reflect an accelerated anabolic demand without adequate replenishment, compromising nitrogen balance in the long term?
Conclusions
The conclusions are overly assertive (“systematically elucidates”, “provides robust theoretical basis”) despite analytical limitations and the inferential nature of the study.
It does not distinguish between confirmed and correlative findings; the claim of an “axis gut microbiota–host amino acid metabolism” is made without experiments proving causality.
Is there physiological or productive evidence that this dose could be extrapolated to other breeds or developmental stages?
References
Lack of specific references on foal digestive physiology and equine microbiota.
Respectfully,
Author Response
Dear Reviewer,
We sincerely appreciate your valuable comments and constructive suggestions. We have carefully addressed each of your comments, and corresponding revisions have been made in the manuscript. All revised sections are highlighted in green in the revised version. The detailed responses are as follows:
Title
(1)Revision of causal wording
The original title implied a causal mechanism. Since the microbial functional analysis in this study is based solely on predictive inference without direct metabolic validation, we revised the title to: “Dietary Proline Supplementation Promotes Growth and Development in Weaned Foals by Modulating Gut Microbial Amino Acid Metabolism”, thereby removing the causal implication.
Abstract
(1)Causal description
In the original abstract, the sentence “furthermore, it enhanced the gut microbiota’s amino acid degradation and metabolic capacity, a finding consistent with the downward trend observed in plasma concentrations of the related amino acids” implied a causal mechanism. Because the microbial functional analysis was only predictive, we revised it to: “Furthermore, functional inference from PICRUSt2 revealed that medium-dose proline supplementation was associated with a higher microbial potential for amino acid degradation and metabolism, in line with the decreasing plasma concentrations of the corresponding amino acids” (Lines 27–30).
(2)Abstract vs. Methods – Sample size consistency
We carefully rechecked and confirmed that our experiment included four groups with seven weaned foals per group, for a total of 28 foals. The sample size has been corrected in the Abstract (Line 11).
(3)Animal-specific information
We have added relevant details regarding the foals’ sex, body weight, and age: “Twenty-eight weaned foals (approximately 5 months of age; body weight: 54.45 ± 11.33 kg; with an equal number of males and females)” (Lines 11–12).
(4)Plasma amino acid decrease
Since plasma amino acids were measured at a single time point, they were not included in the linear mixed model analysis. However, all foals were raised under identical feeding conditions, and therefore the observed decrease in plasma amino acids is attributable to proline supplementation. Accordingly, the abstract was revised to: “Concurrently, supplementation with medium and high doses of proline significantly reduced the plasma concentrations of amino acids such as proline and arginine (P < 0.05).”
Introduction
(1)Rationale for species selection
We have added the following explanation to justify the choice of species: “Crucially, young animals (such as donkeys, horses, pigs, and sheep) often experience marked weaning stress following abrupt weaning, typically manifested by reduced feed intake and diarrhea [11–14]. During this period of rapid growth and development, their demand for amino acids increases substantially. However, fluctuations in feed intake and intestinal dysfunction after weaning often result in insufficient amino acid intake and absorption [15–20]. Therefore, although the body can synthesize proline endogenously, the rate of synthesis may be inadequate to meet the elevated demand associated with stress and rapid growth, leading to an imbalance between supply and demand [5,6]. This imbalance highlights dietary proline supplementation as an important strategy for maintaining normal growth and physiological function in young animals” (Lines 67–77).
(2)Determination of proline supplementation levels
We added the following statement: “The level of proline supplementation was established with reference to our previous findings and prior studies investigating dietary proline supplementation [12]” (Lines 99–101).
Materials and Methods
(1)Lack of metabolic energy data
We agree with the reviewer that data on energy metabolism would strengthen our study. While our present findings highlight significant shifts in nitrogen metabolism and endocrine function, future studies will include assessments of energy status (e.g., key blood metabolites related to energy balance) to provide a more holistic view of the metabolic effects of proline.
(2)Control of type I error
A total of 28 foals were included in the model fitting. The fixed effects accounted for a large portion of the variance in the dependent variables, and the F-values were sufficiently large, indicating that the main patterns in the data were well captured by the model. Detailed molecular and denominator degrees of freedom, F-values, and p-values are provided in the supplementary tables. For post-hoc multiple comparisons, Tukey’s HSD was applied using the multcomp package (v1.4-28), ensuring strict control of type I error (Lines 118–119).
(3)Individual feed intake
To ensure precise dosing, the daily proline supplement for each treatment group was thoroughly mixed into the morning ration of the concentrated feed. The foals were then individually housed in separate stalls for feeding, allowing us to confirm the complete consumption of the supplemented feed. This management practice guaranteed that each foal received its designated daily proline dose. (Lines 107–112).
(4)Microbiota data supplementation and validation
We agree that functional inference from 16S rRNA data has limitations. To directly address this and validate the predicted microbial metabolic functions, our future work will employ a targeted multi-omics approach.⸻
Results
(1)Multiple comparison estimates
Due to the large volume of data, we only presented p-values in the main text. However, the estimates, standard errors , and degrees of freedom for post-hoc comparisons are provided in the Contrasts worksheet of the supplementary tables.
(2)Hepatic enzyme (AST/ALT) discussion
We appreciate the reviewer’s insight. We agree that the original phrasing, especially the use of “in contrast,” might have given the impression that elevated liver enzymes were interpreted as a uniformly positive metabolic outcome, which was not our intention. The manuscript has been thoroughly revised to clearly separate the discussion of liver enzyme increases from BUN results:
“For ALT, levels in all proline-supplemented groups (LG, MG, and HG) were significantly higher than in the control group by day 30 (P < 0.004), although no significant differences were detected among the supplementation groups (P > 0.05). At day 60, estimated marginal means were as follows: HG, 14.00 U/L (95% CI: 11.62–15.05); MG, 13.67 U/L (95% CI: 11.95–15.38); LG, 13.33 U/L (95% CI: 11.62–15.05); and DG, 11.00 U/L (95% CI: 9.28–12.72) (Table S2). BUN levels exhibited a downward trend with increasing proline supplementation. By day 30, BUN concentrations were significantly higher than those of the DG group (P < 0.01) in all supplementation groups except LG (P > 0.16). By day 60, BUN levels had increased in all supplementation groups, with the HG group showing significantly higher values than the other three groups (P < 0.02) (Table S2)” (Lines 359–369).
(3)Microbiota α-diversity and interindividual variability
We fully agree with the reviewer that the observed differences in α-diversity (Chao1, Shannon) were minor and may be influenced by substantial interindividual variability. As the study was designed as an “endpoint” analysis, only post-intervention samples were collected, preventing intra-individual (pre- vs. post-intervention) consistency analysis. This represents a clear limitation of our study. The primary objective was to compare the terminal gut microbiota composition across treatment groups. Despite the lack of baseline data, Adonis/PERMANOVA and other statistical methods accounting for within-group variation revealed significant differences in overall microbial community composition between groups (Figure 7K). This indicates that, although α-diversity changes were minimal, the structure of the microbial community was significantly affected by the treatments. We acknowledge that the absence of rarefaction validation and covariance analysis may affect result robustness, and future studies will incorporate these important analyses
Discussion
(1)Over-interpretation of causal relationships in microbial remodeling
We fully acknowledge the reviewer’s concern regarding the over-interpretation of causality in microbial functional remodeling. To address this, we have revised the corresponding text (Lines 593–597) as follows: “Importantly, the abundance of microbial pathways involved in amino acid metabolism was negatively correlated with the corresponding amino acid concentrations in host plasma, suggesting that remodeling of gut microbial amino acid catabolism may play a key role in regulating systemic amino acid homeostasis.” This wording emphasizes correlation rather than causation, consistent with the predictive nature of the PICRUSt2 analysis.
(2)GH/IGF-1 references
We recognize that the original references may not have perfectly matched the discussion points regarding GH/IGF-1. We have replaced the previous references ( 21–23) with more appropriate citations to ensure logical consistency .
(3)Limitations of PICRUSt2 functional inference
We have added a statement acknowledging the inherent limitations of our functional prediction approach (Lines 614–615): “Limitations exist, including a relatively small sample size and the inherent constraints of functional inference based on 16S rRNA sequencing and PICRUSt2.” This addition clarifies that our microbiota functional predictions are inferred and should be interpreted cautiously.
(4)High-dose proline effects, immune response, and metabolic cost
In response to concerns about the potential adverse effects of high-dose proline supplementation and its impact on immunity and nitrogen metabolism, we have added the following discussion (Lines 570–588):
“Furthermore, proline catabolism may enhance arginine synthesis, which in turn induces arginase expression, activates the urea cycle, and accelerates amino acid catabolism [41–44]. This metabolic shift may explain several observations in the HG group foals: on one hand, the early increase in IgG and IgM levels may be related to enhanced amino acid turnover, providing sufficient substrates for immunoglobulin synthesis; on the other hand, the concomitant elevation of plasma urea nitrogen (BUN) directly reflects increased amino acid catabolism and nitrogen metabolism, suggesting a shift in metabolic state rather than a simple improvement in utilization efficiency [45]. Notably, immunoglobulin synthesis incurs substantial metabolic costs, and the subsequent decline in immunoglobulin levels may reflect a trade-off, wherein resources are preferentially allocated to rapid growth rather than the sustained production of high levels of immune proteins [46]. Overall, for weanling foals, a supplementation dose of 60 mg/kg·d appears to promote growth but may approach the upper limit of the optimal metabolic range. Future studies should incorporate more comprehensive clinical chemistry and histopathological assessments to further evaluate its long-term safety. In contrast, the MG group achieved effective growth while maintaining lower BUN levels, suggesting that this dose may achieve a more favorable balance among substrate supply, nitrogen metabolism, and immune homeostasis.”
Conclusions
Avoiding causal claims and emphasizing predictive correlations
In the revised Conclusions section, we have made extensive modifications to avoid implying causation and to reflect the correlative nature of our findings: All words implying “confirm” or “establish” were replaced with terms such as “indicate,” “suggest,” or “may.” The microbiota functional results are explicitly described as predicted. Direct statements about regulatory axes were removed and replaced with descriptions of “potential interactions.” Future research directions, including causal experiments and studies across other breeds or developmental stages, were highlighted. The revised text reads as follows:
“This study investigated how dietary proline supplementation influences the growth of weaned foals, with a particular focus on the potential interplay between the gut microbiota and host amino acid metabolism. Our results indicate that a moderate dose of proline (40 mg/kg·d) achieved an optimal balance between growth promotion and nitrogen utilization under the experimental conditions. Notably, proline supplementation was associated with predicted remodeling of microbial amino acid catabolic functions, and these inferred microbial changes were inversely correlated with plasma amino acid concentrations. This correlative evidence suggests that microbiota-mediated modulation may be one of the mechanisms through which proline affects growth. Furthermore, our data support the notion that proline contributes to growth by influencing growth-related hormones, immune function, and antioxidant capacity. Collectively, these findings provide initial insights into host–microbiota interactions during weaning and offer a foundational framework for developing nutritional strategies to mitigate weaning stress. Future studies, including causal experiments such as fecal microbiota transplantation and investigations across other breeds and developmental stages, are warranted to confirm the generalizability and mechanistic basis of these observations.”
References
Relevant references have been added or updated to match the revised text.
Kind regards,
Chen Meng

Reviewer 2 Report
Comments and Suggestions for Authors
Thank you for the study. Please, read through the abstract to make necessary correction on methods that stipulated 24 against 28 stipulated in materials and methods.
Can you provide more references for the findings in line 55-56 (add more references to 5,6), also lines 59, 64, 65, 66, 67, 68.
Is there an ethical approval for the study? (Please, indicate). Generally, can you reference protocols used for the various sections in materials and methods (e.g, sample collection and measurement, blood sample and parameter measurements).
Author Response
Dear Reviewer,
We sincerely appreciate your valuable comments and constructive suggestions. We have carefully addressed each of your comments, and corresponding revisions have been made in the manuscript. All revised sections are highlighted in green in the revised version. The detailed responses are as follows:
Comment 1:
The sample size reported in the Abstract is inconsistent with that in the Materials and Methods section.
We carefully rechecked and confirmed that our experiment included four groups with seven weaned foals in each group, for a total of 28 foals. The sample size has been corrected in the Abstract (Line 11).
Comment 2:
Please add supporting references, especially for the statements regarding nonessential amino acids and their metabolic roles in energy metabolism, antioxidation, and immunity.
We have added relevant references to support these statements. Specifically, the literature concerning “the endogenous synthesis rate of certain nonessential amino acids cannot meet the body’s requirements” has been added (Line 57). Additionally, references related to energy metabolism, antioxidation, and immunity have been cited separately (Lines 64–65). The content in Lines 66–68 is now supported by References 11–14.
Comment 3:
Please provide details on the ethical approval, including the approval number and date.
The ethical approval statement has been included in the manuscript (Lines 635–637):
Institutional Review Board Statement: The experimental protocol was approved by the Institutional Animal Care and Use Committee of Xinjiang Agricultural University (Urumqi, China) under approval number 2023020 and approval date 20 April 2023.
We sincerely thank you again for your insightful comments, which have helped us improve the clarity and scientific rigor of our manuscript.
Kind regards,
Chen Meng

Round 2
Reviewer 1 Report
Comments and Suggestions for Authors
Dear authors,
I appreciate the authors’ careful attention to the comments provided in the previous review. After examining the revised version and the response letter, I confirm that all observations have been adequately addressed. The adjustments made to the writing, methodology, and discussion satisfactorily resolve the issues previously noted.
The manuscript can be considered ready for acceptance.
Sincerely,